



# Impact of methane and other precursor emission reductions on surface ozone in Europe: Scenario analysis using the EMEP MSC-W model

Willem E. van Caspel[1], Zbigniew Klimont[2], Chris Heyes[2], and Hilde Fagerli[1]

[1]EMEP MSC-W, Norwegian Meteorological Institute, Oslo, Norway
[2]International Institute for Applied Systems Analysis, Laxenburg, Austria

**Correspondence:** W.E. van Caspel (willemvc@met.no)





**Abstract.**

The impacts of future methane ($CH_4$) and other precursor emission changes are investigated for surface ozone ($O_3$) in the United Nations Economic Commission for Europe (UNECE) region excluding North America and Israel (the "EMEP region", for European Monitoring and Evaluation Programme) for the year 2050. The analysis includes a Current Legislation (CLE) and Maximum Feasible Technical (MFR) reduction scenario, and a scenario that combines MFR reductions with an additional dietary shift that also meets the Paris Agreement objectives with respect to greenhouse gas emissions (LOW). For each scenario, background $CH_4$ concentrations are calculated using a probabilistic Earth System model emulator, and combined with other precursor emissions in a three-dimensional Eulerian chemistry-transport model. While focus is placed on peak season maximum daily 8-hour average (MDA8) $O_3$ concentrations, a range of other indicators for health and vegetation impacts are also discussed. Our analysis show that roughly one-thirds of the total peak season MDA8 reduction achieved between the 2050 CLE and MFR scenarios is attributable to $CH_4$ reductions, resulting predominantly from $CH_4$ emission reductions outside of the EMEP region. The impact of other precursor emission reductions is split nearly evenly between the reductions inside and outside of the EMEP region. However, the relative importance of $CH_4$ and other precursor emission reductions is shown to depend on the choice of $O_3$ indicator, though indicators sensitive to peak $O_3$ show generally consistent results. The analysis also highlights the synergistic impacts of $CH_4$ mitigation as reducing solely $CH_4$ achieves, beyond air quality improvement, nearly two-thirds of the total global warming reduction calculated for the LOW scenario compared to the CLE case.



**Plain language summary**

Methane in the atmosphere contributes to the production of ozone gas, which is an air pollutant as well as a greenhouse gas.
In this study, the impact of reducing methane emissions on surface ozone is investigated for the United Nations Economic
Commission for Europe (UNECE) region excluding North America and Israel (the "EMEP region"), in particular in terms of
its importance in reaching the ozone exposure guideline limits set by the World Health Organization. The relative importance
of reducing emissions of other pollutants that lead to the formation of ozone, such as nitrogen oxides, is also investigation. To
this end, our study employs emission scenarios up to the year 2050, each having different assumptions about future human-
caused emissions. Relative to a scenario where only the already agreed emission reductions are implemented, one-third of
surface ozone reductions in the most ambitious emission reduction scenario are attributable to methane reductions. The other
two-thirds are attributable to emission reductions of other ozone forming pollutants, with reductions inside and outside of the
EMEP region contributing roughly equally.



## 1 Introduction

Surface ozone ($O_3$) is an important source of air pollution, impacting both human and ecosystem health (Lefohn et al., 2018; Monks et al., 2015). In the lower troposphere, the majority of $O_3$ is produced by the photochemical reaction of nitrogen oxides ($NO_x = NO + NO_2$) in carbonaceous-rich environments (Crutzen et al., 1999). The most abundant carbonaceous $O_3$ precursor species is methane ($CH_4$), having a present-day volume mixing ratio of around 1915 parts-per-billion (ppb). Moreover, $CH_4$ mixing ratios are likely to increase further, as anthropogenic $CH_4$ emissions are anticipated to increase in the coming decade

(UNEP, 2021; Saunois et al., 2020; Höglund-Isaksson et al., 2020). In addition to being a source of air pollution, $CH_4$ is also the second most important anthropogenic greenhouse gas (GHG), with its importance as both an air pollutant and global warming agent having received considerable attention in recent years (Mar et al., 2022; Abernethy et al., 2021; Fiore et al., 2008; Dentener et al., 2005).

  In this study, the impact of $CH_4$ and other precursor emissions is investigated for the European Monitoring and Evaluation

Programme (EMEP) region, which includes the member countries of the United Nations Economic Commission for Europe (UNECE) region excluding North America and Israel. Focus is placed on the population-weighted exposure to peak season (April-September) average maximum daily 8-hour mean (MDA8) $O_3$ concentrations, being the health indicator employed by the new World Health Organization (WHO) guidelines (WHO, 2021). The latter recommends a peak season MDA8 exposure limit of 60 µg m$^{-3}$ based on the association between long-term $O_3$ exposure and all-cause mortality, with an interim target of

70 µg m$^{-3}$ for areas where initial exposure is high. To our knowledge, neither the guideline nor interim target values are met in any of the countries within the EMEP region at present. The focus on peak season MDA8 is also motivated by the broader association between the exposure to peak $O_3$ and all-cause mortality (Huangfu and Atkinson, 2020).

  In the current work, the impacts on $O_3$ are investigated for a Current Legislation (CLE), Maximum technical Feasible Reduction (MFR), and MFR with an additional dietary shift and Paris Agreement policy scenario (LOW) up to the year 2050.

The CLE scenario includes the currently agreed upon policies for the abatement of air pollutant and GHG emissions, while the MFR scenario combines the economic activity pathway of the CLE scenario with the full implementation of the best available emission reduction technologies defined in the GAINS (Greenhouse gas – Air pollution Interactions and Synergies) model (Amann et al., 2011). The LOW scenario extends the MFR by including climate policies compatible with the Paris Agreement objectives and an additional shift in agricultural practices, bringing further $CH_4$ and precursor emission reductions. Relative

to the year 2015, global anthropogenic $CH_4$ emissions decline by 35 % and 50 % in the LOW scenario by 2030 and 2050, respectively, making the reductions comparable to those of the Methane Pledge (30 % by 2030, Malley et al., 2023) and Global Methane Assessment (45 % abatement target for 2050, UNEP, 2021).

  The emission scenarios are combined with the Model for the Assessment of Greenhouse-gas Induced Climate Change v7.5.3 (MAGICC7) (Meinshausen et al., 2020, 2011, 2009) to calculate their respective background $CH_4$ concentrations up to the

year 2050. To calculate their impacts on surface $O_3$, the $CH_4$ projections are specified in the three-dimensional Eulerian Chemistry-Transport Model (CTM) developed at the EMEP Meteorological Synthesising Centre – West (hereafter "EMEP model"), where they are combined with the other precursor scenario emissions. The EMEP model has a long history of policy





support and research development (e.g., Jonson et al., 2018; Simpson, 2013; Simpson et al., 2012), with one of its main tasks
being the modeling of transboundary fluxes of air pollutants as part of the UNECE Convention on Long-range Transboundary
Air Pollution (CLRTAP) (Fagerli et al., 2023). In this capacity, the EMEP model has previously been used in support of the
review of the UNECE Gothenburg Protocol (Protocol to Abate Acidification, Eutrophication and Ground-level Ozone).

The emission scenarios and their implementation are described in more detail Sect. 2. The MAGICC7 model is described in
Sect. 3, where it is used to calculate background $CH_4$ concentrations up to the year 2050. Sect. 4 describes the EMEP model
configuration, while also evaluating the baseline configuration against five years of observations across Europe. For the scenario
calculations presented in Sect. 5, the default modeling configuration involves averaging all results over five meteorological
years, while a linear latitudinal $CH_4$ gradient is imposed to capture the effects of inter-hemispheric variations in emissions.
Sect. 5 further combines regional EMEP model simulations with global simulations to quantify the separate impacts of emission
changes inside and outside of the EMEP region. While focus is placed on the peak season MDA8 indicator, scenario results for
a range of other $O_3$ health and vegetation indicators are also presented. The results are discussed and compared against earlier
studies in Sect. 6, followed by a conclusion in Sect. 7.



**Table 1.** Global emission totals for the CLE, MFR, and LOW emission scenarios in units of Tg yr$^{-1}$. Emission totals within the EMEP region, as defined in Sect. 1, are listed in brackets for the years 2015, 2030, and 2050.

| Species | Scenario | 2015 | 2020 | 2025 | 2030 | 2035 | 2040 | 2045 | 2050 |
|---------|----------|------|------|------|------|------|------|------|------|
| NO$_x$ | CLE | 119 (16) | 111 | 106 | 103 (10) | 102 | 103 | 104 | 106 (9) |
| NO$_x$ | MFR | | | | 65 (6) | 52 | 42 | 40 | 38 (4) |
| NO$_x$ | LOW | | | | 62 (6) | 46 | 33 | 29 | 25 (3) |
| NMVOC | CLE | 121 (15) | 120 | 120 | 118 (14) | 119 | 119 | 120 | 121 (14) |
| NMVOC | MFR | | | | 68 (10) | 63 | 59 | 59 | 59 (8) |
| NMVOC | LOW | | | | 63 (9) | 57 | 52 | 50 | 48 (8) |
| CO | CLE | 517 (50) | 474 | 449 | 427 (37) | 418 | 411 | 408 | 405 (43) |
| CO | MFR | | | | 160 (22) | 139 | 123 | 123 | 123 (18) |
| CO | LOW | | | | 149 (22) | 124 | 102 | 96 | 91 (17) |
| CH$_4$ | CLE | 334 (62) | 345 | 360 | 371 (60) | 385 | 401 | 416 | 428 (60) |
| CH$_4$ | MFR | | | | 229 (28) | 224 | 226 | 220 | 210 (22) |
| CH$_4$ | LOW | | | | 219 (27) | 208 | 202 | 195 | 168 (14) |

## 2 Emissions

The emission scenarios were developed using the global version of the GAINS model (Winiwarter et al., 2018; Klimont et al., 2017; Höglund-Isaksson, 2012; Amann et al., 2011), and provided by the EMEP Center for Integrated Assessment Modelling (CIAM) hosted by the Institute for Applied Systems Analysis (IIASA). The scenarios include annual anthropogenic emission totals of CH$_4$, NO$_x$, Non-Methane Volatile Organic Compounds (NMVOC), carbon monoxide (CO), sulphur oxides (SO$_x$), ammonia (NH$_3$), primary fine Particulate Matter (PM$_{2.5}$), and primary coarse PM (PM$_{co}$), as well as carbonaceous fraction of primary PM represented by black carbon (BC) and organic carbon (OC). For the current work, the key emission species are CH$_4$, NO$_x$, CO, and NMVOC, where the latter three affect the lifetime of CH$_4$ by acting as either net sources (NO$_x$) or sinks (CO and NMVOC) of hydroxyl (OH). The latter in turn affects the lifetime of CH$_4$ by loss against oxidation. The global emission totals for the key species are shown in Table 1, along with their respective emissions within the EMEP region for the years 2015, 2030, and 2050.

The emission scenarios span the period from the baseline year 2015 up to 2050 in 5-year intervals, with the MFR and LOW scenarios diverging from the CLE scenario from 2025 onwards. In the EMEP model, natural emissions of soil NO$_x$ are included based on monthly climatological values from the CAMS-GLOB-SOIL v2.4 inventory (Simpson and Darras, 2021), noting that soil NO$_x$ emissions from the application of manures and mineral nitrogen fertilizers on agricultural land are calculated in the GAINS model. Forest fire emissions are included based on the daily Fire INventory from NCAR version 2.5 (FINNv2.5, Wiedinmyer et al., 2023) dataset, derived from fire detections from both the Moderate Resolution Imaging Spectroradiometer (MODIS) and Visible Infrared Imaging Radiometer Suite (VIIRS) satellite instruments.





## 2.1 Emission scenarios

### 2.1.1 CLE scenario

The CLE scenario assumes the implementation and effective enforcement of all currently committed energy and environmental policies affecting emissions of air pollutants and greenhouse gases. CIAM has undertaken a review and update of historical data (up to 2020) driving emissions of all species in the GAINS model, drawing on information from the statistical office of the European Union (EUROSTAT), International Energy Agency (IEA), UN Food and Agriculture Organization (FAO), in addition to data and emissions reported to the Center on Emission Inventories and Projections (CEIP). For the EU27 countries, the energy and agriculture projections are consistent with the objectives of the European Green Deal and 'Fit for 55' package to make the EU carbon neutral by 2050, while also being consistent with the projections used in the EU 3rd Clean Air Outlook (https://environment.ec.europa.eu/topics/air/clean-air-outlook_en, last access: April 2024). For West Balkan, Republic of Moldova, Georgia, and Ukraine, a similar set of modelling tools was used as for the EU, developing a new consistent set of projections. For other world regions, the GAINS model down-scales projections from IEA and FAO (Alexandratos and Bruinsma, 2012; IEA, 2018), considering updated air pollution legislation from national and international sources (e.g., He et al., 2021; Zhang, 2018), including EU legislation and their implementation in consultation with the EU Member States. For the CLE scenario, the socio-economic activity assumptions are similar to that of the Shared Socioeconomic Pathway 2 with an end-of-century radiative forcing of 4.5 W m$^{-2}$ (SSP2-4.5). The SSP2-4.5 scenario describes the 'middle of the road' for future societal development, as described in Meinshausen et al. (2020), O'Neill et al. (2017), and Riahi et al. (2017) for a range of SSP scenarios. For the background $CH_4$ calculations described in Sect. 3, the CLE scenario emissions are therefore combined with GHG emissions (e.g., $CO_2$ and hydrofluorocarbons) from the SSP2-4.5 scenario.

### 2.1.2 MFR mitigation scenario

The MFR mitigation scenario assumes the full implementation of the proven technical mitigation potential as included in the GAINS model for precursor emissions (Amann et al., 2020, 2013; Rafaj et al., 2018) and $CH_4$ (Gomez Sanabria et al., 2022; Höglund-Isaksson et al., 2020; Höglund-Isaksson, 2012). Technologies to abate air pollution precursor emissions include, for example, end of pipe technologies applied in the power, industry, and transport sector, technology change in industry and residential combustion, as well as measures in agriculture addressing emissions from manures and mineral fertilizer application by, for example, improved manure management techniques and construction low emission housing including covered manure stores. The fossil fuel and solvent sector emissions include improved flaring, maintenance, leakage, and distribution control measures, as well as low-solvent product substitutions. Global emissions of $NO_x$, NMVOC, and CO decline by nearly 80 % by 2050 relative to the 2015 baseline, while $CH_4$ emissions fall by 37 %. These reductions are driven by rapid introduction of stringent emission limit values for stationary and mobile sources, strong decline in fossil fuel use, and access to clean energy for cooking. The MFR energy and agricultural activity projections are the same as those of the CLE scenario, with the MFR scenario also being combined with other GHG emissions from the SSP2-4.5 scenario.



### 2.1.3 LOW mitigation scenario

The LOW mitigation scenario extends the MFR by including several additional policies targeting significant transformations in the agricultural sector. This transformation leads to strong reductions of livestock numbers, especially cattle and pigs. The scenario is based in part on the 'Growing Better report 2019' (The Food and Land Use Coalition, 2019) and other studies
addressing healthy dietary requirements (Kanter et al., 2020; Willett et al., 2019), as used in earlier scenarios for global air pollution studies (Amann et al., 2020). While the LOW scenario has the same energy projections as for the CLE for EU27 countries, the rest of the world now includes climate policies compatible with Paris Agreement goals, making the GHG emissions consistent with those of the 'taking the green road' SSP1-2.6 scenario (Riahi et al., 2017; O'Neill et al., 2017). In the LOW scenario, global $CH_4$ emission decline by 34 % and 50 % by 2030 and 2050 relative to the 2015 baseline, respectively.

## 2.2 Model implementation

The annual mean national and sector (e.g., road traffic and agriculture) emission totals are distributed in time using a set of monthly, weekly, daily, and hourly time-factors based on the global and European CAMS-TEMPO datasets described in Guevara et al. (2021, 2020a, b). For the regional EMEP domain discussed in Sect. 4, the native $0.5° \times 0.5°$ scenario emissions are redistributed to the $0.1° \times 0.1°$ spatial distribution of the most recent EMEP reported emissions (2021) for countries within
the EMEP region (EMEP/CEIP, 2023). However, following the approach used for the Gothenburg Protocol review, native $0.1° \times 0.1°$ gridded emissions from CIAM are used for countries located within the West-Balkan and Economic Co-operation and Development, Eastern Europe, Caucasus and Central Asia (EECCA) regions, and for Türkiye. Countries that lie (partially) within the regional modelling domain but that are not part of the EMEP region, such as North African countries, follow the global $0.5° \times 0.5°$ gridded emissions. International shipping emissions also follow the global $0.5° \times 0.5°$ spatial distribution
provided by CIAM for all simulations. We further note that direct emissions of $CH_4$ are not included in the EMEP model, with concentrations instead being specified on an annual mean basis, as discussed in Sect. 4.1.



## 3 Background CH$_4$

Earth System emulators, sometimes known as Reduced Complexity Models (RCMs), have a long history of development as low-cost alternatives to full complexity climate models. RCMs include simplified parameterizations of, for example, ocean heat uptake, GHG effective radiative forcing, and climate feedbacks, to efficiently estimate future change in climate variables such as GHG concentrations and global-mean surface air temperature (GSAT) (Nicholls et al., 2021, 2020). To this end, the MAGICC7 v7.5.3 RCM has been used in the Intergovernmental Panel on Climate Change (IPCC) Sixth Assessment Report (AR6) (Forster et al., 2021), being calibrated to capture the relationship between emissions and GSAT for the AR6 historical temperature assessment (Nicholls et al., 2022). In the current work, the MAGGIC7 model is run using the 5-yearly annual totals from Table 1 linearly interpolated to annual values and combined with their respective SSP GHG scenario emissions.

In the MAGICC7 model, CH$_4$ sinks are represented by loss against OH in the troposphere, loss to the stratosphere, and soil uptake (Meinshausen et al., 2011). Climate sensitivities for these mechanisms arise from, for example, temperature-driven changes in atmospheric composition, changes in the Brewer-Dobson circulation strength, and changing soil properties. CH$_4$ sources are controlled by the separate contributions arising from anthropogenic, natural, and permafrost (Schneider von Deimling et al., 2012) emissions. Natural emissions are estimated by closing the CH$_4$ budget between the years 2015-2023, for which the IIASA emissions are the same for all scenarios, using observed global mean background CH$_4$ concentrations up to the most recent year as reference (1923 ppb by 2023, Lan et al., 2022). With this approach, natural emissions are estimated at 214.9 Tg yr$^{-1}$, falling within the top-down range of 194–267 Tg yr$^{-1}$ reported by Saunois et al. (2020) for the year 2017. The natural emissions are kept constant throughout the simulation period.

A key feature of the MAGICC7 model is that it can be run in a probabilistic mode, where the results of its 600-member ensemble reflect the uncertainties in the parameters controlling future climate change (Nicholls et al., 2022). The initial parameter values controlling the CH$_4$-cycle are the same for each ensemble run, however, with parameters such as the initial lifetime of CH$_4$ (9.95 yr$^{-1}$) and temperature-sensitivity of the loss against OH (0.07 K$^{-1}$) calibrated to match the projections by Holmes et al. (2013) across the range of Representative Concentration Pathway (RCP) scenarios (Meinshausen et al., 2020). As a result, the inter-ensemble variations for the calculated CH$_4$ projections represent the sensitivity of the different CH$_4$ source and sink terms to temperature change. We note that the net land-to-atmosphere CH$_4$ flux from permafrost is found to have a minimal contribution to the simulation results, with the 600-ensemble emissions falling below 4 Tg yr$^{-1}$ by 2050 for all scenarios.

### 3.1 CH$_4$ projections

Fig. 1 shows the CH$_4$ projections calculated for the CLE, MFR, and LOW scenarios, with the shaded regions indicating the the 5th to 95th percentile (5-95 %) range of the 600-ensemble output. Here the CH$_4$ projections for the SSP1-2.6, SSP2-4.5, and SSP5-8.5 scenarios are included for reference, noting that the IIASA scenario projections fall within the range of the optimistic (SSP1-2.6) and pessimistic (SSP5-8.5) scenarios. For the CLE, MFR, and LOW scenarios, the 2050 global mean CH$_4$ concentrations and their 5-95 % range are calculated as 2236 [2166-2299], 1651 [1597-1700], and 1574 [1512-1627] ppb, respectively. For other years, ensemble mean CH$_4$ concentrations are shown in supplementary Table S1.




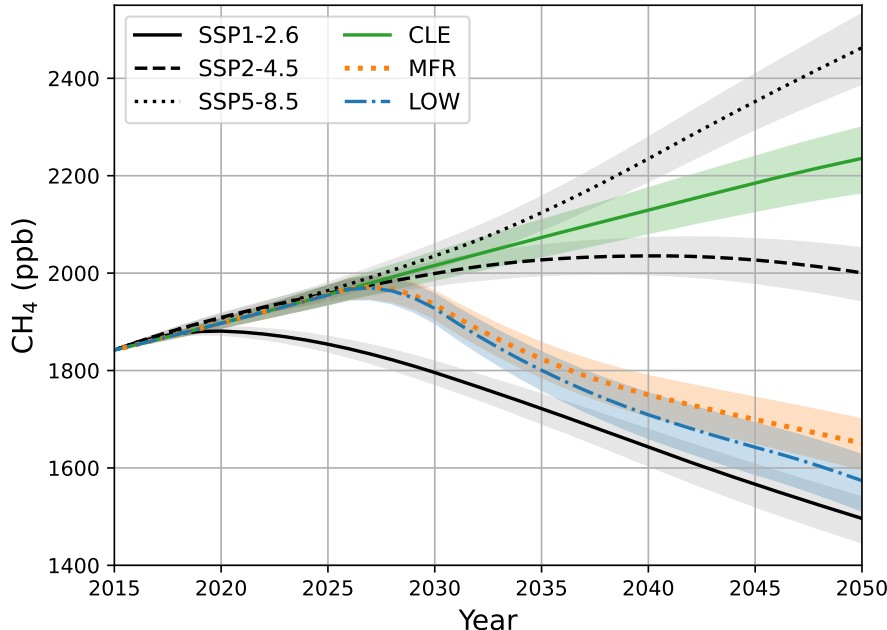

**Figure 1.** Projected background CH$_4$ concentrations up to 2050 for the CLE, MFR, and LOW scenarios described in the text. Projections for the SSP5-8.5, SSP2-4.5, and SSP1-2.6 scenarios are included for reference. The shaded area represents the 600-ensemble 5-95% range.

Fig. 1 shows that the temperature-driven MFR and LOW scenarios uncertainties partly overlap. However, the inter-scenario difference between the CLE and the MFR (and LOW) scenarios far exceeds the temperature-driven uncertainties, with the 2050 ensemble mean difference amounting to 585 ppb. In the current work, the difference between the 2050 CLE and the MFR scenarios represents an important measure of the impact of CH$_4$ emission changes, as this represents the largest inter-scenario concentration change. While both scenarios have a 5-95 % range of approximately 100 ppb by 2050, the 5-95 % interval of

the difference between the 2050 CLE and MFR scenarios amounts to 571-598 ppb. Thus illustrating that ensemble members with a comparatively high CH$_4$ concentration in the CLE scenario also have a comparatively high concentration in the MFR scenario.

Diagnostic simulations for a scenario where CH$_4$ emissions follow the LOW scenario while all other emissions follow those of the CLE scenario (LOW-CH$_4$) are also performed. This hypothetical scenario thereby reflects a situation where CH$_4$ emis-

sions are reduced strongly, while no further abatement policies are implemented for the other emissions. In reality, however, CH$_4$ reductions might also lead to a reduction in other co-emitted species. The resulting 2050 LOW-CH$_4$ concentration of 1440 [1392-1484] ppb is comparable to that of the LOW scenario, although lower by 134 ppb (-8.5 %) due to higher emissions of other lifetime-affecting precursor species. The LOW-CH$_4$ scenario thereby illustrates that the difference in CH$_4$ concentrations between the 2050 CLE and LOW scenarios (and corollary MFR) is primarily driven by the difference in the direct emissions

of CH$_4$, and to a lesser extent by the difference in other precursor and GHG emissions. We further note that continuing the





$CH_4$ projections into 2055 with the emissions fixed to that of 2050 leads to an additional change in the ensemble-mean concentrations of 38 ppb (1.7 %), -45 ppb (-2.7 %), and -70 ppb (-4.45 %) for the CLE, MFR, and LOW scenarios, respectively. The latter illustrates that the $CH_4$ source and sink terms have not yet reached equilibrium by 2050, owing to the relatively long lifetime of $CH_4$.





## 4 EMEP model description

The current work uses EMEP model version rv5.0, as described in more detail by EMEP MSC-W (2023) and others (e.g., Ge et al., 2024; van Caspel et al., 2023; Stadtler et al., 2018; Simpson et al., 2012). The model employs 20 vertical hybrid pressure-$\sigma$ levels for the regional $0.1° \times 0.1°$ EMEP modelling domain (30°N-82°N, 30°W-90°E), and 19 vertical levels for the global $0.5° \times 0.5°$ modelling domain. Both the regional and global grids use 3-hourly meteorological data derived from the ECMWF Integrated Forecasting System (IFS) cycle 40r1 model (ECMWF, 2014). 3-hourly IFS $O_3$ concentrations are specified at the model top (100 hPa) boundary condition, while output surface concentrations are adjusted to an equivalent altitude of 3 meters. Hydrogen gas ($H_2$) is specified with a fixed global concentration of 500 ppb. For the lateral boundary conditions (LBCs) in the regional simulations, 6-hourly output fields from global simulations are used, with each of the global simulations employing a spin-up period of six months. Diagnostic simulations find that the choice of LBC time-resolution has a negligibly small impact on the simulation results, while choosing 6-hourly over 3-hourly LBCs saves considerable computation time. Output fields from the global model are also used as initial conditions for the regional runs. The geographical region spanned by the regional EMEP modelling domain contains the EMEP region, but also parts of North Africa and Asia, whose emissions are consistently treated as Rest-Of-World (ROW) between the global and regional simulations. For reference, the EMEP region as represented in the regional EMEP modelling domain is shown in supplementary Fig. S1.

### 4.1 CH$_4$ implementation

In the EMEP model, global mean CH$_4$ concentrations are specified at the start of each run and remain unchanged over the course of the simulation. However, observed CH$_4$ concentrations display a marked latitudinal gradient, primarily due to the presence of large natural and anthropogenic emission sources in the Northern Hemisphere. The latitudinal gradient can be described by its two leading Empirical Orthogonal Functions (EOFs), or principal components (Meinshausen et al., 2017). The first EOF (EOF1) represents a nearly linear North-South gradient, while the second EOF (EOF2) represents a local northern mid-latitude maximum of ~10 ppb. EOF1 has a pre-industrial North to South pole gradient of around 40-50 ppb, and of around 90 ppb for the year 2014 (Meinshausen et al., 2017). To capture the main characteristics of the latitudinal gradient, the contribution of EOF1 is included in the EMEP model by specifying

$$\mathrm{CH_4}'(\phi, \mathrm{CH_4}) = \mathrm{CH_4}\left(1 + \frac{0.025\phi}{90}\right), \tag{1}$$

where CH$_4$ represents the global mean background concentration and $\phi$ is latitude in degrees. For pre-industrial (808 ppb) and the year 2015 (1834 ppb) global mean CH$_4$ concentrations, Eq. 1 yields latitudinal gradients of 40 ppb and 92 ppb, respectively, consistent with those described in Meinshausen et al. (2017). By applying Eq. 1 also for the projected CH$_4$ concentrations, an approach similar to that of Meinshausen et al. (2020) is followed, by effectively using EOF1 to extrapolate the latitudinal gradient into the future based on anthropogenic CH$_4$ emissions.



**Table 2.** EMEP model configurations for the scenario analysis discussed in Sect. 5. The $CH_4$ concentrations refer to the 2050 global mean values calculated in Sect. 3. Each of the scenarios is simulated for the five meteorological years between 2013-2017.

| Experiment long-name | Short-name | ROW emis | EMEP region emis | $CH_4$ (ppb) |
|---|---|---|---|---|
| Baseline 2015 | bs15_bs15ch4 | 2015 baseline | 2015 baseline | 1834 |
| Baseline 2015 to ROW 2050 CLE emis | rowcle50_bs15ch4 | 2050 CLE | 2015 baseline | 1834 |
| Baseline 2015 to global 2050 CLE emis | cle50_bs15ch4 | 2050 CLE | 2050 CLE | 1834 |
| Global 2050 CLE | cle50_cle50ch4 | 2050 CLE | 2050 CLE | 2236 |
| 2050 CLE to ROW 2050 MFR emis | rowmfr50_cle50ch4 | 2050 MFR | 2050 CLE | 2236 |
| 2050 CLE to global 2050 MFR emis | mfr50_cle50ch4 | 2050 MFR | 2050 MFR | 2236 |
| Global 2050 MFR | mfr50_mfr50ch4 | 2050 MFR | 2050 MFR | 1651 |
| 2050 MFR to ROW 2050 LOW emis | rowlow50_mfr50ch4 | 2050 LOW | 2050 MFR | 1651 |
| 2050 MFR to global 2050 LOW emis | low50_mfr50ch4 | 2050 LOW | 2050 LOW | 1651 |
| Global 2050 LOW | low50_low50ch4 | 2050 LOW | 2050 LOW | 1574 |

## 4.2 Scenario configurations

To simulate the effects of precursor emission changes inside and outside of the EMEP region, regional simulations are combined with LBCs from the global model configuration. Simulations where only background $CH_4$ concentrations are changed serve to isolate the impact of global $CH_4$ change. Since $CH_4$ is a globally well-mixed gas, and since the concentration changes are the result of anthropogenic $CH_4$ emission changes, the impact of the total global mean $CH_4$ change is split into its EMEP region and ROW contributions based on the $CH_4$ emission changes within these respective regions. This approach is supported by the surface $O_3$ response being effectively linear in the range of $CH_4$ concentrations relevant to the current work, as discussed in Sect. 6.1. An overview of the scenario simulations is shown in Table 2, noting that each of the configurations is simulated for each of the five meteorological years between 2013-2017 for both the regional and global setups, as discussed below.

## 4.3 Baseline evaluation against observation

The efficacy of the EMEP model to simulate peak season MDA8 is evaluated by comparing the baseline configuration to surface observations. To this end, the baseline 2015 configuration is used to perform simulations for the 2013-2017 meteorological years, and compared against surface observations from the EBAS database (Tørseth et al., 2012). While the emissions are fixed to that of the year 2015, inter-annual variability in the emissions is generally small. The 56 EBAS stations are located within the European part of the EMEP region (as shown in supplementary Fig. S2), and are selected from all available stations based on the requirement that they each measure peak season MDA8 for each of the five meteorological years. For MDA8, data availability guidelines stipulate that for each 8-hourly mean 75% of the hourly values must be present, while at least 75% of the eight hour averages must be present in a day to assign a maximum daily 8-hour mean (EU, 2008). Data availability guidelines





similar to those for annual mean $O_3$ are also adopted, requiring that at least 90 % of the days between April-September have MDA8 measurements available to assign a peak season average.

Fig. 2a compares the five-year average modelled and observed peak season MDA8 values at each of the 56 stations. A clear relationship between the modelled and observed values is present, having a Pearson correlation coefficient (r) of 0.87. The normalized mean bias (NMB) amounts to 4.2 %, indicating that the model has a slight tendency to overestimate. Fig. 2b shows the annual averages across all 56 stations, illustrating that the total inter-annual variability for both model and measurements corresponds to around 4-5 µg m$^{-3}$. The difference between the total annual average modelled and observed concentrations is greatest for the year 2014, amounting to 6.9 µg m$^{-3}$ (8.2 %), while being as low as 0.6 µg m$^{-3}$ (0.7 %) for the year 2013. The difference in the five-year average measured (84.9 µg m$^{-3}$) and modelled (88.5 µg m$^{-3}$) concentrations follows that of the NMB (3.6 µg m$^{-3}$, or 4.2 %). In Fig. 2c, annual averages across all stations within Sweden, Germany, Spain, the United Kingdom, and Poland are shown, illustrating that the model generally captures the observed variability between high and low $O_3$ years also at regional scales. Observed concentrations were the lowest in 2017, except for in Spain, as also reproduced by the model. The observed differences between the highest (2015) and lowest year (2017) can be as large as 12.7 µg m$^{-3}$ (16.80 %), for example for Poland. The modelled inter-annual variability in the different regions is approximately equal to, or sometimes smaller than (e.g., Poland, Spain), the observed variability. For Poland, the difference between the highest and lowest modelled year amounts to 7.6 µg m$^{-3}$ (8.51 %), being lower by 5.1 µg m$^{-3}$ than the observed maximum variability.

Overall, the EMEP model displays good agreement with observations across the five meteorological years, while highlighting that inter-annual peak season MDA8 variability can be on the order of 10-15 % on regional scales, and around 5 % across Europe. To reduce the effects of meteorological variability, each of the scenarios listed in Table 2 is therefore simulated for the years 2013-2017, with the results presented in the following representing five-year averages.



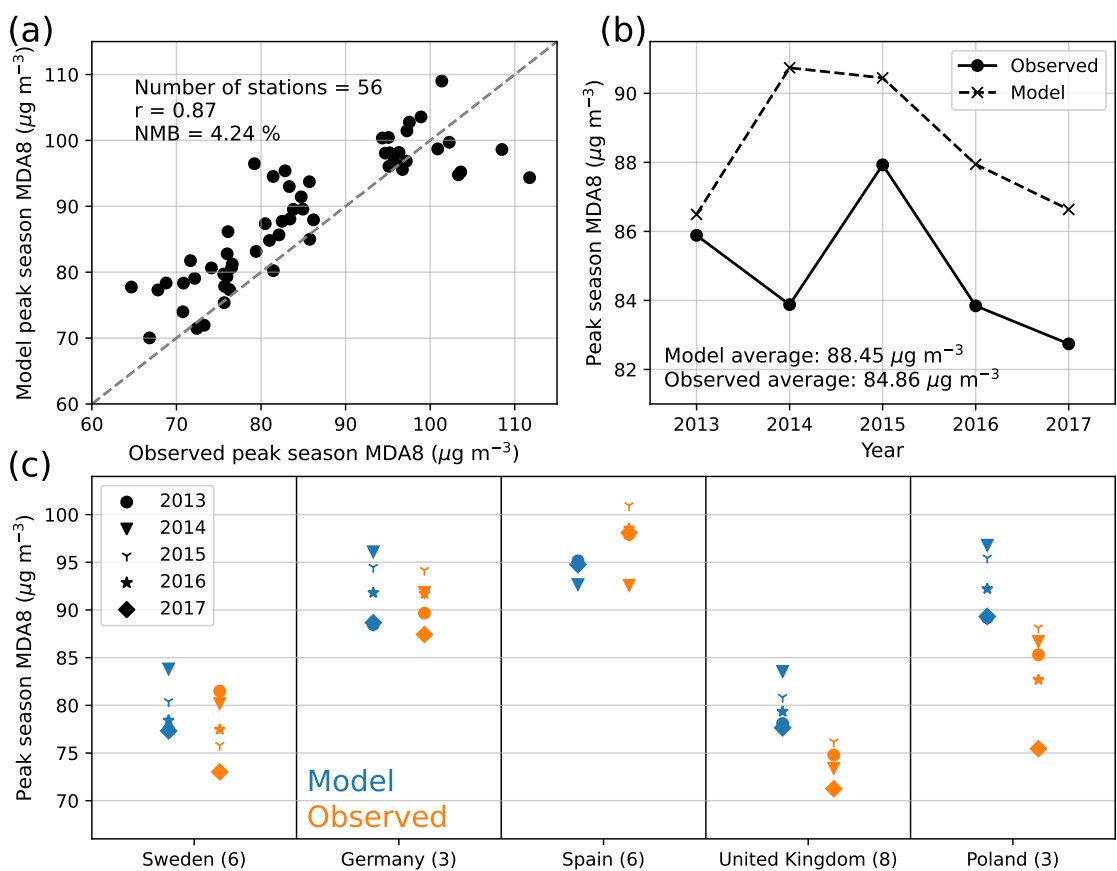

**Figure 2.** Modelled versus observed peak season MDA8 across Europe. Panel (a) shows five-year averaged values at each of the 56 stations, while panel (b) compares the annual values averaged over all stations. Panel (c) shows the yearly averages for Sweden, Germany, Spain, the United Kingdom, and Poland, with the number in brackets indicating the number of stations in each of the countries.



## 5 Results

While the focus in this section lies on peak season MDA8, results for other $O_3$ indicators are included in the supplementary
material, as referred to in the text. In addition, the following discusses a number of weighted averaging approaches for both
health and vegetation $O_3$ indicators, with the different population and crop-area maps shown for reference in supplementary
Fig. S3.

### 5.1 EMEP region peak season MDA8

Fig. 3 shows a so-called cascade-plot of the EMEP region population weighted peak season MDA8 changes between the
2015 baseline and the 2050 CLE, MFR, and LOW scenarios. Here the population weighting is calculated using the Global
Human Settlement Layer (GHSL) population distribution for the year 2015 (Schiavina et al., 2023), aggregated from its native
3 arc-second resolution to the regional EMEP grid, remaining unchanged for all scenarios. In Fig. 3, the impacts arising from
$NO_x$, CO, and NMVOC precursor emissions changes and from $CH_4$ are shown as separate cascade-steps. In the cascades,
the separate contributions arising from EMEP region and ROW emission changes are also highlighted, calculated using the
model configurations described in Table 2. For example, the difference between the 'bs15_bs15ch4' and 'rowcle50_bs15ch4'
simulations yields the change due to 2050 CLE precursor emission changes in the ROW region relative to the 2015 baseline,
while the difference between the 'cle50_bs15ch4' and 'cle50_cle50ch4' simulations yields the change due to background $CH_4$
changes. As noted in Sect 4, the impact of global $CH_4$ emission changes is split into its EMEP and ROW region contributions
based on the emission changes within these respective regions. In effect, the cascade-plot summarizes the impact of each of
successive precursor and $CH_4$ change from the 2015 baseline down to the 2050 LOW scenario.

Fig. 3 shows that average peak season MDA8 concentrations are reduced from 93.3 to 90.3 µg m$^{-3}$ between the 2015
baseline and 2050 CLE scenarios, resulting largely from a decrease in precursor emissions in the EMEP region (-4.8 µg m$^{-3}$)
and to a lesser extent in the ROW region (-1.4 µg m$^{-3}$). However, these reductions are partially offset by an increase of 3.2
µg m$^{-3}$ arising from increased background $CH_4$ concentrations, being almost entirely the result of increased $CH_4$ emissions
in the ROW region. Going from the 2050 CLE to 2050 MFR scenario, the net reduction from 90.3 to 76.8 µg m$^{-3}$ (-15 %) is
split into three nearly equal parts arising from EMEP region precursor reductions, ROW precursor reductions, and background
$CH_4$ reductions. Further, the 2050 LOW scenario differs relatively little from the MFR, with roughly half of the change from
76.8 to 73.4 µg m$^{-3}$ arising from further precursor emission reductions within the EMEP region. Cascade-plots for the annual
$O_3$ mean, SOMO35, and POD$_3$IAM$_{WH}$ indicators discussed in Sect. 5.2, are shown in supplementary Fig. S4-S6.

### 5.1.1 Geographical distribution

To illustrate the impact of geographical location on the $O_3$ changes resulting from precursor and $CH_4$ emission changes, the
difference in peak season MDA8 between the 2050 CLE and LOW scenarios is shown across the regional EMEP modelling
domain in Fig. 4. Here the 2050 CLE to LOW impacts are calculated by combining the results from the 2050 CLE to 2050 MFR
simulations with the 2050 MFR to 2050 LOW simulations described in Table 2. In Fig. 4a, the change in peak season MDA8





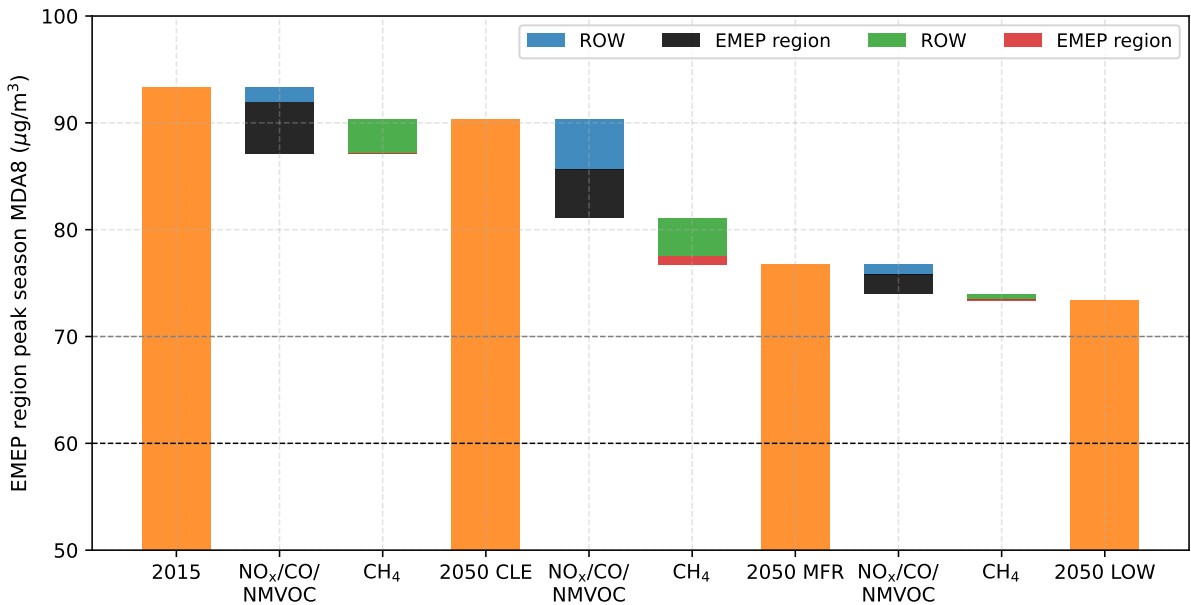

**Figure 3.** Cascade plot of the population weighted EMEP region average peak season MDA8 scenario changes arising from $NO_x$, CO, and NMVOC emission changes within the EMEP (black) and ROW (blue) regions, and from background $CH_4$ changes arising from EMEP (red) and ROW (green) region emission changes. The black and grey dashed horizontal line denote guideline and interim WHO target values, respectively. Note that the y-axis starts at 50 µg m$^{-3}$.

resulting from the change to ROW LOW emissions is shown. As expected, the ROW LOW impacts are most pronounced in the ROW countries within the regional modelling domain (e.g., North African countries). Nevertheless, countries along the Southern border of the EMEP region as well as along the Western coast of Europe also see reductions ranging from 5-15 µg m$^{-3}$. The reductions along the Western coast of Europe are likely the result of emission reductions in North America, with the associated $O_3$ perturbations carried over the Atlantic ocean by the prevailing Westerlies. Fig. 4b shows that the impact

of regional LOW emissions is largely centered on the EMEP region, ranging from approximately 5 µg m$^{-3}$ in Western Europe to 15 µg m$^{-3}$ in the West-Balkan and EECCA countries. While local in nature, the impact of emission reductions in both the EMEP and ROW regions can lead to increases of as much as 30 µg m$^{-3}$ in large urban areas (as highlighted in Fig. 4b for Almaty, Kazakhstan), due to reductions in the titration effect of $NO_x$. The impact of background $CH_4$ reductions from 2236 to 1574 ppb is shown in Fig. 4c, with the latitudinal gradient arising from the latitudinal variations in insolation. The resulting

peak season MDA8 reductions amount to around 5 µg m$^{-3}$ across the EMEP region.

Fig. 4d shows the results for the full 2050 LOW scenario, illustrating that peak season MDA8 concentrations fall below 60 µg m$^{-3}$ over parts of Scandinavia, while ranging from 80 to 90 µg m$^{-3}$ over Northern Italy and Kazakhstan. In Central Europe, concentrations typically range from 60-70 µg m$^{-3}$, highlighting that the WHO exposure guideline of 60 µg m$^{-3}$ is not met anywhere in the majority of EMEP countries. However, the interim target of 70 µg m$^{-3}$ is reached in a number of



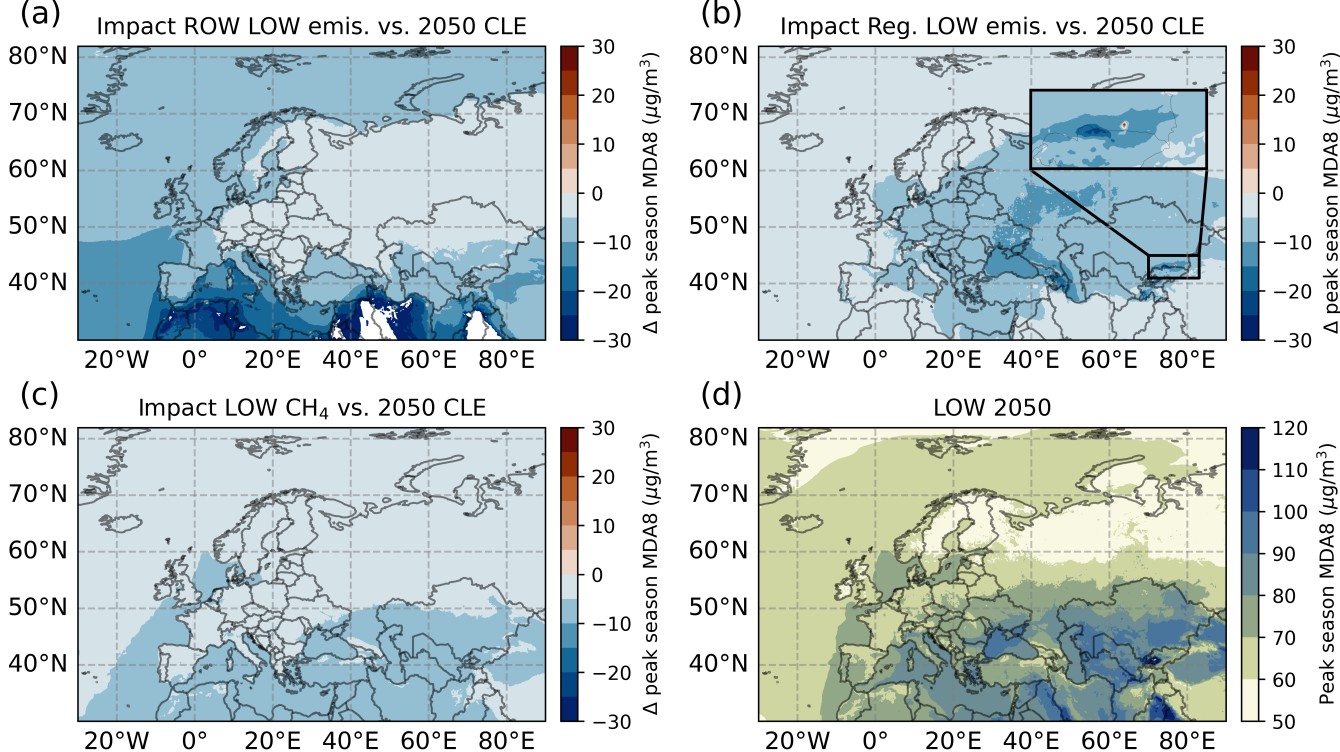

**Figure 4.** Reductions in peak season MDA8 achieved by 2050 ROW LOW (panel a) and EMEP region (panel b) precursor emission changes relative to the 2050 CLE scenario. Panel (b) also highlights the simulation results for Almaty, Kazakhstan. Panel (c) shows the reductions arising from the background $CH_4$ change from 2236 to 1574 ppb, while panel (d) shows the peak season MDA8 as simulated for the full 2050 LOW scenario. Note the difference in color-scale for panel (d).

Western European countries, such as the Netherlands, France, Germany, and the United Kingdom. The population weighted LOW scenario concentrations for each of the individual countries in the EMEP region are shown in supplementary Fig. S7, along with their 2015 baseline and 2050 CLE and MFR concentrations. In addition, supplementary Fig. S8 follows that of Fig. 4, but instead compares the impacts of the LOW scenario against the 2015 baseline. For the latter, the impact of regional emission reductions is comparatively higher, while that of $CH_4$ changes is comparatively lower, consistent with the results

shown in Fig. 3.

**5.2 Other $O_3$ indicators**

This section serves to provide reference to earlier studies by showing the scenario results for a range of other health and vegetation $O_3$ indicators. For example, earlier works have investigated the impact of precursor and $CH_4$ emission changes on (area weighted) annual mean surface $O_3$ ("$O_3$ mean") concentrations (Turnock et al., 2018; Jonson et al., 2018), while the

Sum of Ozone Means Over 35 ppb (SOMO35), 4th highest annual MDA8, and summertime (JJA) average daily maximum $O_3$



concentrations have been used for health impact studies (Fleming et al., 2018). Furthermore, JJA average $O_3$ concentrations were used for the study of the climate-impact on surface $O_3$ by Colette et al. (2015), as will discussed in more detail in Sect. 6. For the impacts on vegetation, the growing-season accumulated Phyto-toxic Ozone Dose ($POD_Y$) uptake over a certain threshold value $Y$ (nmole $O_3$ $m^{-2}$ $s^{-1}$) can induce reductions in crop and semi-natural biomass (Emberson, 2020; Mills et al.,

2018). To this end, the Integrated Assessment Modelling (IAM) vegetation-type specific $POD_Y$ indicators ($POD_Y IAM$) serve as simplified risk assessment indicators for use in CTMs such as the EMEP model (Simpson et al., 2012, 2007), as also described in the UNECE 'Mapping Manual' (UNECE, 2017). The $POD_3 IAM_{WH}$ indicator represents the cumulative growing-season (∼90 days) stomatal $O_3$ uptake for a generic temperate or boreal crop, being largely based on wheat (WH), and is used as an indicator for wheat yield loss (Pandey et al., 2023; Mills et al., 2018). In addition, the $POD_1 IAM_{DF}$ indicator is used in

the risk assessment of reductions in annual living deciduous forest (DF) biomass growth (UNECE, 2017), having a ∼180 day growing season at 50°N.

Table 3 shows the absolute and percentage change scenario results across the range of $O_3$ indicators, for an extended range of (constructed) scenarios. For example, the '2015 Base to 2050 MFR' scenario is constructed using the differences between the '2015 Base to 2050 CLE' and the '2050 CLE to 2050 MFR' scenarios described in Table 2, while the '2050 CLE to 2050

LOW' scenario is constructed using the differences between the '2050 CLE to 2050 MFR' and '2050 MFR to 2050 LOW' scenarios (as described in Sect. 5.1.1). Likewise, the '2015 Base to 2050 LOW' scenario is constructed using the differences between the '2015 Base to 2050 CLE', '2050 CLE to 2050 MFR', and '2050 MFR to 2050 LOW' scenarios. Note that for peak season MDA8, the absolute numbers shown for the '2015 Base to 2050 CLE', '2050 CLE to 2050 MFR', and '2050 MFR to 2050 LOW' scenarios correspond to those shown in Fig. 3. Furthermore, since the relative importance of $CH_4$ emission

changes inside the EMEP region is small, Table 3 only includes the impact of global $CH_4$ changes. While health-related $O_3$ indicators are shown as population weighted averages, the $POD_Y$ indicators are shown as their respective vegetation-area weighted averages (i.e., average values per square meter of vegetation, as illustrated in supplementary Fig. S3).

Table 3 illustrates that different uses of threshold values and time and length of averaging or accumulation periods leads to differences in the relative importance of precursor and $CH_4$ emission changes. For example, indicators most sensitive to

$O_3$ concentrations during its peak photochemical production period (peak season MDA8, JJA $O_3$ max, JJA $O_3$ mean, and 4th highest MDA8) are most strongly impacted by regional precursor emission reductions, especially when compared against the 2015 baseline scenario. In contrast, regional emission reductions are much less important for annual $O_3$, due to the competing effects of local wintertime $NO_x$ titration. The importance of ROW emissions is, broadly speaking, proportional to the length of the averaging or accumulation period, while also being most relevant to the 2050 scenarios (i.e., 2050 CLE and MFR). For

indicators employing a threshold value, the percentage-change impacts are proportional to the height of the threshold relative to the baseline (or background) value, which effectively determines the degree of which the natural background is filtered out. For example, the total percentage-change reduction from the 2015 baseline to 2050 LOW scenarios for the SOMO35, $POD_3 IAM_{WH}$, and $POD_1 IAM_{DF}$ indicators amounts to 63.5 %, 36.9 %, and 32.1 %, respectively. While already implied in Fig. 3, Table 3 also shows that the impact of $CH_4$ emission reductions is most important relative to the 2050 CLE scenario and

less so when compared against the 2015 baseline. However, $O_3$ mean is an exception to the latter, with $CH_4$ having the largest



**Table 3.** Absolute and percentage change (in brackets) scenario impacts across the EMEP region. Changes resulting from precursor emission changes in the EMEP (reg.) and ROW regions, and from global $CH_4$ changes, are shown relative to the scenario starting points. End values correspond to the weighted averages at each of the scenario end-points.

| Scenario | | 2015 Base to 2050 CLE | 2015 Base to 2050 MFR | 2015 Base to 2050 LOW | 2050 CLE to 2050 MFR | 2050 CLE to 2050 LOW | 2050 MFR to 2050 LOW |
|---|---|---|---|---|---|---|---|
| PS MDA8[a] | Reg. emis | -4.8 (-5.1 %) | -9.3 (-10.0 %) | -11.2 (-12.0 %) | -4.5 (-5.0 %) | -6.4 (-7.1 %) | -1.9 (-2.4 %) |
| | ROW emis | -1.4 (-1.5 %) | -6.1 (-6.5 %) | -7.0 (-7.5 %) | -4.7 (-5.2 %) | -5.6 (-6.3 %) | -1.0 (-1.2 %) |
| | $CH_4$ | 3.1 (3.4 %) | -1.2 (-1.2 %) | -1.7 (-1.9 %) | -4.3 (-4.8 %) | -4.9 (-5.4 %) | -0.6 (-0.8 %) |
| | End value | 90.3 | 76.8 | 73.4 | 76.8 | 73.4 | 73.4 |
| $O_3$ mean[a] | Reg. emis | 1.3 (2.0 %) | -0.2 (-0.3 %) | -0.9 (-1.4 %) | -1.5 (-2.3 %) | -2.1 (-3.3 %) | -0.7 (-1.2 %) |
| | ROW emis | -1.1 (-1.8 %) | -5.5 (-8.8 %) | -6.6 (-10.6 %) | -4.4 (-6.7 %) | -5.5 (-8.5 %) | -1.1 (-2.0 %) |
| | $CH_4$ | 2.2 (3.4 %) | -0.8 (-1.3 %) | -1.2 (-2.0 %) | -3.0 (-4.6 %) | -3.4 (-5.2 %) | -0.4 (-0.7 %) |
| | End value | 65.0 | 56.2 | 54.0 | 56.2 | 54.0 | 54.0 |
| 4th MDA8[a] | Reg. emis | -9.3 (-6.4 %) | -15.2 (-10.4 %) | -18.0 (-12.3 %) | -5.8 (-4.2 %) | -8.7 (-6.2 %) | -2.8 (-2.3 %) |
| | ROW emis | -1.0 (-0.7 %) | -4.7 (-3.2 %) | -5.6 (-3.8 %) | -3.7 (-2.7 %) | -4.6 (-3.3 %) | -0.9 (-0.7 %) |
| | $CH_4$ | 3.1 (2.1 %) | -1.2 (-0.8 %) | -1.7 (-1.2 %) | -4.3 (-3.1 %) | -4.9 (-3.5 %) | -0.6 (-0.5 %) |
| | End value | 139.1 | 125.2 | 120.9 | 125.2 | 120.9 | 120.9 |
| JJA $O_3$ max[d] | Reg. emis | -7.4 (-7.1 %) | -13.3 (-12.7 %) | -15.7 (-14.9 %) | -5.9 (-5.9 %) | -8.2 (-8.3 %) | -2.3 (-2.7 %) |
| | ROW emis | -1.2 (-1.1 %) | -5.2 (-5.0 %) | -5.9 (-5.6 %) | -4.0 (-4.0 %) | -4.7 (-4.7 %) | -0.7 (-0.8 %) |
| | $CH_4$ | 3.7 (3.6 %) | -1.3 (-1.3 %) | -2.0 (-1.9 %) | -5.1 (-5.1 %) | -5.8 (-5.8 %) | -0.7 (-0.8 %) |
| | End value | 99.9 | 84.9 | 81.2 | 84.9 | 81.2 | 81.2 |
| JJA $O_3$ mean[a] | Reg. emis | -3.6 (-4.9 %) | -6.9 (-9.3 %) | -8.3 (-11.2 %) | -3.2 (-4.5 %) | -4.7 (-6.5 %) | -1.4 (-2.3 %) |
| | ROW emis | -0.9 (-1.3 %) | -4.4 (-6.0 %) | -5.0 (-6.7 %) | -3.5 (-4.8 %) | -4.0 (-5.6 %) | -0.6 (-0.9 %) |
| | $CH_4$ | 2.9 (3.9 %) | -1.0 (-1.4 %) | -1.6 (-2.2 %) | -3.9 (-5.5 %) | -4.5 (-6.2 %) | -0.5 (-0.9 %) |
| | End value | 72.0 | 61.4 | 58.8 | 61.4 | 58.8 | 58.8 |
| SOMO35[b] | Reg. emis | -427 (-13.2 %) | -891 (-27.5 %) | -1049 (-32.4 %) | -464 (-15.4 %) | -622 (-20.7 %) | -158 (-10.5 %) |
| | ROW emis | -162 (-5.0 %) | -809 (-25.0 %) | -919 (-28.3 %) | -647 (-21.5 %) | -756 (-25.2 %) | -109 (-7.3 %) |
| | $CH_4$ | 355 (11.0 %) | -44 (-1.4 %) | -89 (-2.8 %) | -399 (-13.3 %) | -445 (-14.8 %) | -45 (-3.0 %) |
| | End value | 3009 | 1498 | 1185 | 1498 | 1185 | 1185 |
| POD$_3$IAM$_{WH}$[c] | Reg. emis | -1.7 (-10.1 %) | -3.0 (-18.2 %) | -3.5 (-21.2 %) | -1.3 (-8.7 %) | -1.8 (-11.9 %) | -0.5 (-4.4 %) |
| | ROW emis | -0.4 (-2.5 %) | -1.9 (-11.3 %) | -2.1 (-12.8 %) | -1.4 (-9.4 %) | -1.7 (-11.0 %) | -0.2 (-2.2 %) |
| | $CH_4$ | 1.0 (5.8 %) | -0.3 (-1.9 %) | -0.5 (-3.0 %) | -1.3 (-8.3 %) | -1.4 (-9.4 %) | -0.2 (-1.5 %) |
| | End value | 15.4 | 11.3 | 10.4 | 11.3 | 10.4 | 10.4 |
| POD$_1$IAM$_{DF}$[c] | Reg. emis | -2.4 (-10.1 %) | -4.3 (-18.2 %) | -4.9 (-20.9 %) | -1.9 (-8.8 %) | -2.5 (-11.7 %) | -0.6 (-3.7 %) |
| | ROW emis | -0.5 (-2.1 %) | -1.8 (-7.8 %) | -2.1 (-8.9 %) | -1.3 (-6.2 %) | -1.6 (-7.4 %) | -0.3 (-1.5 %) |
| | $CH_4$ | 1.0 (4.2 %) | -0.4 (-1.5 %) | -0.5 (-2.3 %) | -1.3 (-6.2 %) | -1.5 (-7.0 %) | -0.2 (-1.1 %) |
| | End value | 21.5 | 16.9 | 15.9 | 16.9 | 15.9 | 15.9 |

[a]Population weighted EMEP region average in µg m$^{-3}$. [b]Population weighted EMEP region average in ppb day$^{-1}$. [c]Crop-area weighted EMEP region average in mmol m$^{-2}$.
[d]Population weighted average converted from ppb to µg m$^{-3}$ using the standard-atmosphere $O_3$ conversion factor of 1.96.





percentage-change impact from the 2015 baseline to 2050 CLE scenario. Furthermore, $CH_4$ reductions contribute roughly one thirds of the total reductions for each of the peak $O_3$ indicators for the 2050 CLE to 2050 MFR scenario, although this is closer to one-fourths for SOMO35 (26.4 %).

For the population weighted $O_3$ indicators (i.e., all except those for vegetation), the corresponding area weighted averages are shown in supplementary Table S2. While the results are generally consistent between the two weighted averaging approaches, indicators sensitive to peak $O_3$ concentrations are comparatively less impacted by regional precursor emission changes when calculated as area weighted averages. However, the area weighted impacts of regional precursor emission changes are considerably larger for annual $O_3$ mean, since $NO_x$ titration effects in urban areas is weighted less heavily. For example, reducing regional emissions between the 2015 baseline to 2050 MFR scenarios sees a population weighted $O_3$ mean reduction of 0.2 µg m$^{-3}$ (0.3 %), while the corresponding area weighted reduction amounts to 4.5 µg m$^{-3}$ (6.7 %).





## 6 Discussion

In the current setup, the EMEP model is unable to capture the effects of future climate change on surface $O_3$ concentrations. This effect, often described as the $O_3$ climate penalty (e.g., Fu and Tian, 2019; Rasmussen et al., 2013), can affect surface $O_3$ for example through climate-change induced changes in water vapour concentrations and biogenic VOC emissions. For
European land surfaces, Colette et al. (2015) estimated the 95 % confidence interval of the mid-century (2041-2070) surface JJA $O_3$ mean climate penalty to range from 0.44-0.64 ppb, based on an ensemble of 25 chemistry-climate model simulations. Compared to the JJA $O_3$ mean changes between the 2050 CLE and MFR scenarios shown in Table 3, amounting to 10.6 µg m$^{-3}$ (or 5.3 ppb using the standard-atmosphere $O_3$ conversion factor of 1.96), the impact of the climate penalty on the results of the current work is expected to be small. Other climate-uncertainties relate to the calculated $CH_4$ projections, with
natural soil emissions estimated to increase by $22.8 \pm 3.6$ Tg $CH_4$ yr$^{-1}$ by the year 2100 in the SSP5-8.5 scenario (Guo et al., 2023). Such changes are nevertheless small relative to the total natural emissions, estimated at 210 Tg yr$^{-1}$ in Sect. 3, also considering that the scenario analysis of the current work goes up to the year 2050.

While constructing emission data sets based on a wide variety of information is by itself challenging (e.g., de Meij et al., 2024; Thunis et al., 2022), the emission scenarios employed in the current work are also inherently based on a number of socio-
economic activity projections. In practice, the reliable quantification of the uncertainty on the input parameters to the GAINS model is itself considered the most uncertain element of the analysis (Amann et al., 2011). In this light, the emission scenarios arguably represent the largest source of uncertainty for the current work, which is unavoidable and not directly quantifiable. Nevertheless, the GAINS model by design attempts to minimize the impact of uncertainties on policy-relevant model output, to increase the robustness (i.e., the priorities and control needs between countries, sectors and pollutants do not significantly
change due to uncertainties in the model elements) of the emission control strategies (Amann et al., 2011).

### 6.1 $O_3$ production efficiency of $CH_4$

The $CH_4$ oxidation reaction that leads the production of $O_3$ depends on the availability of $NO_x$ and OH (Crutzen et al., 1999). OH is produced through the photolysis of $O_3$ and subsequent reaction of $O(1D)$ with water vapour ($H_2O$), with the majority of surface $O_3$ being produced by the photolysis of $NO_x$ in carbonaceous-rich environments. In addition, CO and
VOCs (including $CH_4$) are net sinks of OH, creating a non-linear relationship between their atmospheric abundance and the $O_3$ production efficiency (OPE) of $CH_4$ (Isaksen et al., 2014). In the current work, the OPE is taken as the capacity of $CH_4$ to produce surface $O_3$ in the EMEP region. To investigate the impact of OPE on the calculated $O_3$ response, diagnostic simulations are performed where background $CH_4$ concentrations are varied between 850 to 2600 ppb in 250 ppb steps, using both the 2015 baseline and the 2050 CLE and LOW emission scenarios as the source of background precursor emissions.
The resulting $CH_4$ impacts on peak season MDA8 are shown in Fig. 5, noting that the starting point of 850 ppb corresponds roughly to pre-industrial $CH_4$ concentrations. For simplicity, the simulations shown here are only calculated for the 2015 meteorological year.





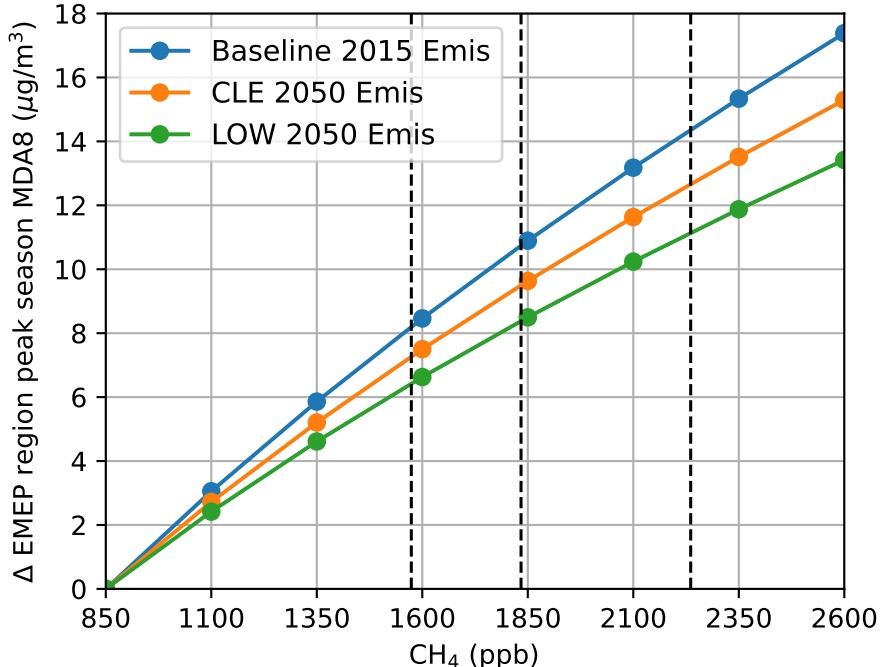

**Figure 5.** Change in EMEP region population weighted peak season MDA8 for background $CH_4$ concentrations ranging from 850 to 2600 ppb in 250 ppb intervals, relative to peak season MDA8 concentrations at 850 ppb $CH_4$. The impacts are calculated for the baseline 2015 and the 2050 CLE and LOW emission scenarios defined in Table 2, with the dashed vertical lines marking their calculated 2050 background $CH_4$ concentrations (1574, 1834, and 2236 ppb, respectively).

Fig. 5 illustrates that the OPE is highest in the 2015 baseline scenario, when EMEP region $NO_x$ emissions are also highest (Table 1). Regional $NO_x$ emissions are reduced strongly in the 2050 CLE scenario, while other emissions change relatively
little. As a result, the OPE is a factor of 0.88 smaller relative to the 2015 baseline across the range of $CH_4$ concentrations. Similarly, the OPE in the 2050 LOW scenario is a factor of 0.88 lower than that of the 2050 CLE scenario, and by a factor of 0.78 relative to the 2015 baseline scenario. In the analysis of Sect. 5, the largest $CH_4$ differences occur between the 2050 CLE and LOW scenarios, ranging from 2236 to 1574 ppb. For this concentration interval, the decrease in peak season MDA8 amounts to 5.4 and 4.7 µg m$^{-3}$ when calculated from the 2050 CLE and LOW precursor emission scenarios, respectively. Since
this represents a comparatively small difference, the $CH_4$ impacts described in Sect. 5 are robust with respect to the choice of emission scenario in which the $CH_4$ concentrations are reduced. Fig. 5 furthermore illustrates that the peak season MDA8 response is approximately linear in the range of $CH_4$ concentrations relevant to the current work, supporting the approach of splitting the $O_3$ impacts based on the separate emission changes within the EMEP and ROW regions. Another corollary is that the contribution of anthropogenic background $CH_4$ to total peak season MDA8 amounts to approximately 10.7 (11.5 %), 12.7
(14.0 %), and 6.4 (8.7 %) µg m$^{-3}$ in the 2015 baseline, 2050 CLE, and 2050 LOW scenarios, respectively. Here the percentage



contributions are based on the scenario totals shown in Fig. 3 and Table 3. Recognizing that the MFR and LOW precursor emission scenarios are nearly identical except for $CH_4$ emissions, the anthropogenic $CH_4$ contribution calculated for the 2050 MFR scenario (1651 ppb) amounts to 7.0 µg m$^{-3}$ (9.1 %).

## 6.2 Comparison to previous studies

Comparing to the results of the current work to earlier studies can be challenging, for example due to differences in source-receptor area definitions, model configuration, weighted averaging approach, and emission scenarios. Nevertheless, the EMEP region total peak season MDA8 exposure reduction by 15 % between the 2050 CLE and MFR scenarios is consistent with the 16 % reduction found by Belis and Van Dingenen (2023) across the entire UNECE region. However, in our calculations the total 2050 MFR anthropogenic $CH_4$ contribution amounts to 7.0 µg m$^{-3}$ (or 3.5 ppb using the standard-atmosphere $O_3$

conversion factor of 1.96), which is lower than their estimate of $\sim$ 5 ppb (based on their Fig. S4). This can largely be reconciled considering that our estimate was calculated with the 2050 LOW scenario as the source of background precursor emissions, while theirs is based on $O_3$ sensitivities calculated from a 2010 baseline emission scenario. When using the 2015 baseline emission scenario as the source of background precursor emissions in our calculations, the total 2050 MFR anthropogenic $CH_4$ contribution amount to 9.0 µg m$^{-3}$, or 4.5 ppb, which is more comparable.

In the work of Turnock et al. (2018), the box-model described in Holmes et al. (2013) is used to estimate the 2050 CLE and MFR $CH_4$ concentrations to amount to 2361 and 1420 ppb, respectively. They further estimate the 2050 CLE increase in $CH_4$ to contribute 1.6 ppb to annual mean area weighted $O_3$ across Europe relative to a 2010 baseline concentration of 1798 ppb, based on the parameterized response of 14 models. While the latter is higher than our estimate of 1.1 ppb for the EMEP region (Table S3, using the standard-atmosphere $O_3$ conversion factor of 1.96), our results find a more comparable contribution of 1.4

ppb when the response is calculated as the European area weighted average following the land-area definition of Turnock et al. (2018). However, in our results the 2050 CLE and MFR ensemble mean $CH_4$ concentrations amount to 2236 and 1651 ppb, respectively, with the total difference between the CLE and MFR scenarios therefore being 403 ppb (or 43 %) less than that of Turnock et al. (2018). While this may in part be due to their MFR scenario diverging from the CLE from 2020 rather than 2025 onwards, it nevertheless highlights the importance of the methodology used to estimate $CH_4$ concentrations, as the cumulative

difference between scenarios can quickly diverge. The difference in $CH_4$ estimates also has implications for the impact of the 2050 MFR emissions relative to the baseline, which in our analysis (-183 ppb) is around half that determined by Turnock et al. (2018) (-378 ppb).

## 6.3 Air-pollution and global warming co-benefits

While a detailed discussion is beyond the scope of the current work, the global mean temperature change relative to the

reference period of 1986-2005, as calculated for the 600-ensemble mean and 5-95 % range using the MAGICC7 model in Sect. 3, amounts to 2.21 [1.61-2.94], 2.02 [1.45-2.74], and 1.92 [1.33-2.67] degrees K for the 2050 CLE, MFR, and LOW scenarios, respectively. In the LOW-$CH_4$ scenario, where $CH_4$ emissions follow the LOW scenario while all other emissions follow that of the CLE, this change amounts to 2.03 [1.47-2.74] degrees K. Thus illustrating that around two-thirds of the



global warming reduction between the 2050 CLE (SSP2-4.5 GHGs) and LOW (SSP1-2.6 GHGs) scenarios can be achieved by

solely reducing $CH_4$ emissions. Note that all emissions follow those of the corresponding SSP scenarios before 2015, as the

IIASA scenarios only start from 2015 onwards.





# 7 Conclusion

This work investigates the impact of $CH_4$ and other precursor emissions on surface $O_3$ concentrations in the EMEP region for the CLE, MFR, and LOW emission scenarios up to the year 2050. In the CLE scenario, background $CH_4$ concentrations are

projected to increase by 402 ppb (22 %) relative to 2015 baseline concentrations, while they are reduced by 183 ppb (-10 %) in the MFR scenario. The difference between the 2050 MFR and CLE scenario therefore amounts to 585 ppb (or 26.1 % less in the MFR compared to the CLE scenario), while the LOW scenario achieves a modest further 77 ppb reduction. The MFR $CH_4$ reductions lead to a peak season MDA8 exposure reduction of 4.3 µg m$^{-3}$ (4.8 %) relative to the 2050 CLE case, contributing around one-thirds of the total peak season MDA8 reduction (13.5 µg m$^{-3}$, or 15 %). The other two-thirds are

split almost equally between the impact of other precursor emission ($NO_x$, CO, NMVOC) reductions in the EMEP and ROW regions, respectively. As for $CH_4$, the impact of further abatement policies for the other precursor emissions is comparatively small in the LOW scenario. While the impacts of background $CH_4$ changes are relatively small when measured against the 2015 baseline, the increase in $CH_4$ in the CLE scenario nevertheless offsets the peak season MDA8 reductions achieved by precursor emissions reductions in the EMEP region almost entirely.

In terms of the total reductions, the 2050 MFR scenario brings the EMEP region average peak season MDA8 exposure down from 90.3 to 76.8 µg m$^{-3}$ relative to the CLE, against a 2015 baseline exposure of 93.3 µg m$^{-3}$. Nevertheless, in the MFR scenario the majority of countries in the EMEP region (38 out of 49) are projected to stay above the interim WHO exposure target of 70 µg m$^{-3}$. While the more stringent emission policies of the LOW scenario reduce the number of countries to 31, it still highlights the difficulties in reaching WHO guideline values, given that the MFR and LOW scenarios already include

optimistic global and regional emission reductions. For some mostly Northern European countries, the LOW scenario brings surface peak season MDA8 concentrations close to the 60 µg m$^{-3}$ WHO guideline value, even though none of their population weighted averages reach below this limit.

    While the current work focuses on the peak season MDA8 indicator, the scenario results are also discussed for a range of other health and vegetation $O_3$ indicators. These results find that the relative importance of $CH_4$ and other precursor emission

reductions depends on the choice of indicator, and to some extent on the spatial averaging approach between area and population weighted. The percentage change scenario impacts can vary greatly, being mostly dependent on the degree to which a threshold value applies. For example, the total reduction between the CLE and MFR scenarios for the SOMO35 health indicator and the POD$_3$IAM$_{WH}$ vegetation indicator amounts to 50 % and 26 %, respectively, compared to a 15 % total reduction for peak season MDA8. Nevertheless, $O_3$ indicators emphasizing peak $O_3$ concentrations yield results largely consistent with

those for peak season MDA8 in terms of the relative importance of the different emission changes. For these indicators, reducing precursor emissions other than $CH_4$ within the EMEP region, or Europe, further has the largest potential to reduce the impact of surface $O_3$ exposure relative to the 2015 baseline.

    The EMEP modelling configuration described in the current work also serves to define the setup for future scenario work performed at MSC-W, both for the EMEP region as well as other source-receptor regions. Future work can also go out to quan-

tifying the risks for mortality and vegetation yield loss based on their associations with the range of $O_3$ indicators. In addition,



the results of the current work contribute to the discussion surrounding the second revision of the Gothenburg Protocol, for which the impact of $CH_4$ on surface $O_3$ plays a prominent role. The current work also highlights that reducing $CH_4$ emissions achieves considerable global warming reductions, with solely reducing $CH_4$ emissions achieving roughly two-thirds of the global warming reduction potential between the full 2050 CLE (SSP2-4.5 GHGs) and LOW (SSP1-2.6 GHGs) scenarios.

However, the global warming and surface air quality reductions are almost entirely the result of, and can only be achieved by, $CH_4$ emission reductions outside of the EMEP region.



*Code and data availability.* The EMEP MSC-W CTM version rv5.0 is available from https://zenodo.org/record/8431553 (EMEP MSC-W, 2023) (last access February 2024). The EMEP input files and output data fields specific to the current work, in addition to the Python scripts used for the data analysis and figure creation, are available from van Caspel et al. (2024). The latter data repository also contains the Python

scripts used to create the MAGICC7 input and run files. The MAGICC7 model, 600-ensemble probabilistic distribution, and SSP emission scenarios can be downloaded after registration from https://magicc.org/download/magicc7 (last access February 2024). The EBAS data are available from https://ebas.nilu.no/ (last access February 2024).

*Author contributions.* HF and WEvC conceptualized the work, while WEvC performed the simulations, did the analysis, and wrote the manuscript. ZK contributed to the text of Sect. 2. CH created the emission scenario files. All authors reviewed the manuscript before

submission.

*Competing interests.* The authors declare that no competing interests are present.

*Acknowledgements.* This work has been funded by the EMEP Trust Fund. IT infrastructure in general was available through the Norwegian Meteorological Institute (MET Norway). Some computations were performed on resources provided by UNINETT Sigma2 - the National Infrastructure for High Performance Computing and Data Storage in Norway (grant NN2890k and NS9005k). The CPU time made available

by ECMWF has been critical for both for generation of meteorology used as input for the EMEP MSC-W model as well as the calculations presented in the current work.

The EBAS database has largely been funded by the UN-ECE CLRTAP (EMEP), AMAP and through NILU internal resources. Specific developments have been possible due to projects like EUSAAR (EU-FP5)(EBAS web interface), EBAS-Online (Norwegian Research Council INFRA) (upgrading of database platform) and HTAP (European Commission DG-ENV) (import and export routines to build a secondary

repository in support of www.htap.org, last access April 2024). A large number of specific projects have supported development of data and meta data reporting schemes in dialog with data providers (EU, CREATE, ACTRIS and others). For a complete list of programmes and projects for which EBAS serves as a database, please consult the information box in the Framework filter of the web interface. These are all highly acknowledged for their support.

We acknowledge the use of Pyaerocom (Gliss et al., 2020, https://pyaerocom.met.no/) (last access February 2024) for creating the co-

located model and measurement data files used in the comparison of EBAS data against the simulations. We also acknowledge the use of the MAGICC7 Python wrapper Pymagicc version 2.1.4, available from https://github.com/openscm/pymagicc (last access February 2024).



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
