# Peer review of "Impact of methane and other precursor emission reductions on surface ozone in Europe: Scenario analysis using the EMEP MSC-W model"

_EGUsphere, 2024_

## Author Comment (AC1)

We wish to thank the reviewer (RC1) for their thoughtful and helpful comments. They have undoubtedly helped to improve the quality of the manuscript. Please find below our point-by-point answers, as highlighted in blue.

General comments

An interesting study, overall clear and well written. Uses a current legislation and maximum feasible reduction scenario to study ozone changes in Europe, splitting out impacts from methane changes and precursor emissions reductions, illustrating the role of these on air quality and also how use of different impact metrics can affect their relative importance. Fig 3 is really interesting, clear and a useful summary of the main results of the study.

**Specific comments**

Section 2.1.2: could add comment on the timeline of introduction of emissions reduction technologies and the choice of divergence of scenarios from 2025 onward

We have added a sentence in the introduction of Sect. 2 motivating to our choice of the scenarios diverging from 2025 onward (i.e., the first year where annual emission totals differ is 2026). In short, while the MFR scenario includes the rapid introduction of stringent emission control measures, the time for policy makers to decide upon emission control strategies takes years at best. Bearing that in mind, the scenarios differing from 2025 onward (first year with different emissions being 2026) is an optimistic target.

Section 2.1.3: A different background GHG scenario is used for LOW. To what extent does this limit being able to compare the ozone impact of methane reductions directly to the CLE and MFR scenarios? This effect is likely to be limited, due to the scenarios having the same NOx, CO, NMVOCs, but a comment on this would be helpful. Maybe related to the discussion in lines 407-411.

While we are not able to incorporate the effects of GHG changes (or climate change) in the EMEP model directly (except for the CH4 chemistry), we performed diagnostic simulations where the LOW-scenario CH4 projections calculated with the MAGICC7 model used SSP2-4.5 rather than SSP1-2.6 GHGs. These simulations found that the change to SSP2-4.5 had very little impact on the calculated CH4 projections (< 4 ppb by 2050 for all ensemble members). With CH4 arguably being the GHG most relevant to (surface) ozone, we expect the GHG scenario (for gasses other than CH4) would also have only a very small effect on the ozone concentrations simulated using the EMEP model.

We have added a sentence describing the diagnostic GHG MAGICC7 simulation to Section 3.

177/Fig 1: SSP3-7.0 is the most extreme/'pessimistic' scenario in terms of methane, so this scenario could be added (or replace SSP5-8.5) to show the full range of methane trajectories in the SSPs

We have performed simulations where the MAGICC7 scenario calculations, as shown in Fig. 1 of the manuscript, include also those for SSP3-7.0. These results were consistent with those reported by Meinshausen et al. (2020), with SSP3-7.0 having the highest methane concentrations by 2100. However, as can also be seen from Fig. 11c from Meinshausen et al. (2020), the SSP3-7.0 and SSP5-8.5 CH4 concentrations are nearly identical by the year 2050, with their concentrations starting to diverge from roughly 2060 onward. Instead of including SSP3-7.0 as a separate curve in Fig. 1, we have therefore decided to include reference to the small 2050 difference between SSP3-7.0 and SSP5-8.5 in the text of Sect. 3.1.

197-199: Mention that this is as expected since the methane emissions (i.e. input source terms) have not stabilized by 2050, and the system would not be expected to be in equilibrium.

This has now been added to the text.

248: How many stations are excluded based on the data availability requirements? And does this affect particular regions/countries? E.g. Fig S2 shows no stations in Italy, south-eastern Europe

There are around 110-120 stations with ozone measurements available for each of the years between 2013-2017. With our requirement that each of the selected stations should measure peak season MDA8 for each of the five years, 56 stations remain, as shown in Fig. 2 of the manuscript.

While there is some degree of mismatch between which stations are operational in each of the years, we find that the most impactful selection criterion is the requirement of having MDA8 measurements available for 90% of the days between April and September (peak season) in order to assign a peak season average. Relaxing this requirement to 75% of the days increases the number of stations with a peak season MDA8 measurement for each of the five years to 87. Reducing the requirement even further to 50% increases the number of stations to 95. Since the change in stations going from 75% to 50% data availability is modest, and since the number of stations with the 75% availability requirement represents ~75-80% of the total number of stations, we will below focus on the stations that are available with 75% daily MDA8 availability.

As shown in Fig. R1 below, following panels (a) and (b) of Fig. 2 of the manuscript, the 5-annual average for each of the stations (panel a) and the average across all stations for each of the years (panel b) does not show any particular signs of change. That is, the conclusions drawn from this figure are the same as those drawn from Fig. 2 of the manuscript, with the NMB and Pearson correlation coefficient being practically unchanged (3.75 % to 4.13 %, and 0.85 to 0.85, respectively). Fig. R1 thus illustrates that our stringent data availability requirement for peak season MDA8 does not impact our conclusions with regards to model performance, while being consistent with the data requirements used in previous studies for the calculation of an annual mean ozone.

[Figure]

Figure R1. Modeled versus observed peak season MDA8 across Europe. Panel (a) shows five-year averaged values at each of the 56 stations, while panel (b) compares the annual values averaged over all stations.

Fig. R2 below shows the spatial distribution of the 87 stations (following Fig. S2 of the manuscript). Fig. R2 illustrates that the lack of observational data in South-Eastern Europe and minimal coverage in Italy (though there is now one station available on the island of Lampedusa), is not necessarily the result of our selection criteria, but rather from the low or inconsistent number of observations available in these regions from the EBAS data base.

A sentence highlighting the discussion surrounding Fig. R1 and R2 has been added to Section 4.3 of the revised manuscript.

[Figure]

Figure R2. Geographical locations of the 87 EBAS stations shown in Fig. R1, having peak season MDA8 measurements available for each of the years between 2013-2017.

252: Mention (perhaps in discussion) how the normalized mean bias in the model compared to obs affects the results and conclusions shown – since the results are differences/anomalies with respect to CLE or 2015 baselines this is limited but a sentence could be added to comment on this. For example, it is likely to affect the metrics based on a threshold more?

In case that the model shows a positive bias because the contribution of natural emissions is overestimated, the % change impact of anthropogenic reductions will be smaller. This effect is enhanced for indicators employing a threshold, since such indicators in effect take out a portion of the contribution of natural emissions. However, it could also be argued that the model shows a positive bias because it overestimates the impact of anthropogenic emissions, in which case the positive bias does not necessarily lead to an underestimation of the % change impacts. However, investigating how these effects compete is not straight forward, and beyond the scope of the current work.

Nevertheless, we have added a sentence to the conclusion, reflecting that our results regarding the number of countries reaching WHO (interim) guideline values might be regarded as somewhat of an upper estimate, since the EMEP model shows a positive bias.

309: Would expect the spatial distribution of reduction in Fig 4c to not just be due to insolation changes but also the spatially different chemical environments? Which would affect the ozone impact of methane changes. Is this based on analysis or just a proposed explanation

Earlier global analysis for the impact of methane changes (not shown, but also based on the global spin-up simulations performed for the current work) does indicate that insolation variations contribute to a large extent to the latitudinal gradient in the ozone impacts of CH4, as the gradient also extends over, e.g., the oceans. However, in the comparatively regional European domain, the impact of variations in chemical environments can't simply be excluded altogether. We have therefore added that the latitudinal gradient "probably to a large extent arising from" rather than "arising from" latitudinal variations in insolation.

348-349: I think this is an important result, and is hard to see from looking at the table with so many values (which are useful for reference). Perhaps it could be colour coded by percentage change, which would e.g. show clearly which indicators show a stronger impact, and how threshold/time/averaging makes a difference.

To our knowledge, color coding text is not currently an option in ACP, even though we understand the reviewer's concern. We hope that in the final type setting phase, when Table 3 receives its final formatting (typically featuring more closely spaced numbers), it will be easier to make out the differences.

379: Related to climate uncertainty in CH4 projections – wetland emissions increases (and climate feedbacks) are likely to have a much larger effect than natural soil emissions, and here natural emissions are assumed to be constant (as standard in CMIP6) e.g. Kleinen et al 2021 (https://iopscience.iop.org/article/10.1088/1748-9326/ac1814/pdf), Zhang et al 2017 (https://www.pnas.org/doi/epdf/10.1073/pnas.1618765114)

Thank you for the interesting references. Going by the cited work of Guo et al. (2023), wetlands are referred to as a type of soil, with the 'terrestrial soil CH4 emissions' referring to the net CH4 flux resulting from emissions (predominantly from wetlands) and uptake by soil (upland soils being the primary sink). For consistency with the terminology used by Guo et al. (2023), we have changed the wording on line 379 from 'natural soil emissions' to 'terrestrial soil emissions'. We note that the paragraph of line 379 has also been adjusted in response to RC2.

**Section 6.1:**

391-396: To what extent are these processes included in the model? Does it have interactive OH

The model (EMEP model) includes fully interactive chemistry, and thus also interactive OH concentrations. We have adjusted the text in Section 6.1 to specifically state that the simulations discussed in this section are performed with the EMEP model, while also describing the chemical mechanism of the EMEP model in more detail in Section 4.

Also, are each of these points run for a single year? Does it require spin up for 150ppb jumps in CH4 concentration?

These points are indeed run for a single year (2015, as noted in the text), though with each having a 6-month spin-up period following the model configuration description in Section 4. In the revised manuscript we have added this information also to the text of Section 6.1, for clarity.

When concluding that Fig 5 shows that the peak season MDA8 response is linear with CH4 (411), to what extent are non-linear processes captured in these experiments? E.g. large influence of methane burden on OH concentration, affecting the oxidative capacity and therefore ozone production.

To address this point, as well as the points raised above, we have added additional information to Section 4 (describing the EMEP model configuration), describing the chemical mechanism (EmChem19) employed by the EMEP model in greater detail. We hope that with this, the role of methane in modulating OH concentrations in our scenario simulations is clarified.

Fig 5: It would be helpful to label the dashed lines on the figure e.g. CLE 2050, MFR 2050 etc. The ordering of labels in the caption is also slightly confusing: 'The impacts are calculated for the baseline 2015 and the 2050 CLE and LOW emission … (1574, 1834, and 2236 ppb, respectively)', not clear which number corresponds to which scenario, if following the text it should be 1834, 1574, and 2236 ppb, respectively(?)

The wording in the figure caption was indeed wrong, which has been fixed in the revised manuscript. In addition, we have added a vertical dashed line marking the 2050 MFR background CH4 concentration. We also tried adding labels to the vertical dashed lines, or color-coding them, but found it to make the figure rather messy. We therefore hope that, with the added vertical line and correctly formatted figure caption, it is now more clear what the vertical dashed lines represent.

407-411 'In the analysis... concentrations are reduced' : I found this a bit hard to understand, I think it could be rearranged/reworded to put a summary sentence to introduce what the main point being made is. E.g.: The decrease in peak season MDA8 was found to be most strongly influenced by the background CH4 concentration, rather than the precursor emissions scenario, with a 5.4 and 4.7 ug m-3 decrease with CLE and LOW precursor emissions respectively, for the same CH4 decrease (from 2236 to 1574ppb). Do the 5.4 and 4.7 numbers correspond to any rows in Table 3?

The numbers relate to the 4.9 ug m-3 reduction from CH4 in the 2050 CLE to 2050 LOW scenario in Table 3, which given the scenario configuration, is calculated based on a mixture of background 2050 MFR and LOW emissions.

The suggested sentence indeed clarifies the purpose of the text, and we have largely adopted its form in the revised manuscript. The updated text also refers to the relevant part of Table 3.

450: SSP scenarios only start in 2015, when they all branch from historical emissions? So they should all be the same before 2015 anyway. Think this sentence can be deleted or replaced with something along the lines of 'Note that emissions/scenarios are the same before 2015.'

Thank you for pointing out this oversight, indeed the SSP GHGs are all the same before 2015. We have now simply omitted this sentence.

462-464: 'the increase in CH4 in the CLE scenario nevertheless offsets the peak season MDA8 reductions achieved by precursor emissions reductions in the EMEP region almost entirely.' Does this refer to +3.4% for CH4 and –5.1% for precursor emissions? Suggest to weaken/delete 'almost entirely'. Could also add a sentence linking to impacts/policy e.g. this highlights the need for simultaneous reductions in both CH4 and precursor emissions.

This indeed referred to +3.4 % for CH4 and -5.1% for other precursor emissions, which we agree do not 'almost entirely' offset each other. The latter has been changed in the text, while we have also updated the conclusion to better reflect the (policy relevant) results.

489: suggest to rephrase 'global warming reduction potential' to avoid confusion with GWP. E.g. possible temperature reduction

We have rephrased to 'possible temperature reduction'.

**Technical corrections**

23: typo - 'is also investigation'

Changed.

33: please add reference and year for 1915ppb CH4 mixing ratio

A reference is given in Section 3 for the historical globally averaged annual background CH4 measurements. We have now also included this reference in the introduction/line 33.

119: typo - 'construction low'

Changed to 'the construction of low'

190: suggest 'might' -> 'likely'

Changed.

294: n.b. POD3IAMWH not defined before this, probably fine since the sentence points to sect. 5.2

Indeed, we have refrained from defining the acronym here to keep its definition and discussion contained to Sect. 5.2, but mention it on line 294 regardless for those readers familiar with the metric.

327: typo 'will discussed'

Changed to 'will be discussed'.

405: minor comment – maybe easier to read a % reduction in OPE rather than 'a factor of XX lower'

The percentage changes have been added as additional information in brackets following the 'factor' statements (e.g., factor of 0.88 (12%) smaller). Hopefully this also removes some ambiguity as to what a factor of 0.88 reduction means (maybe some might interpret it as a 88% reduction).

We wish to thank the second reviewer (RC2) for their helpful comments, which helped to improve the quality of the manuscript. Please find below our point-by-point answers, highlighted in blue. The ordering of the comments follows those of the annotated .pdf file, skipping over the first four empty comments in the abstract.

The reviewer further noted that "I think, the implications of the results for policy making could be extended a bit more in the Discussions." In light of this remark, we have reworded and extended the conclusion section to better highlight the implications for policy.

1. perhaps: VOC-rich

VOCs indeed have a more clear definition as photochemical ozone precursors; we have replaced "carbonaceous-rich" in the text at three different places.

2. add reference please

Done.

3. Are those emissions in Tg(N) yr-1, Tg(NO) yr-1 or Tg(NOx) yr-1. If Tg(NOx) yr-1, what molecular weight is assigned to NOx and what is the NO2:NO ratio assumed in Nox?

The NOx emissions are in Tg(NOx) yr-1, having a molecular weight of 14 + 2 * 16 = 46 g/mol. We note that for use in the MAGICC7 model, the NOx emissions are scaled to Tg(N) yr-1 using a factor of 14 / (14 + 2 * 16). For the MAGICC7 model the NO2:NO emission ratio is not relevant.

In the EMEP model simulations, the NO2:NO ratio is assumed 95:5, with the exception for ship-plumes in pristine air environments. Information on the MW of NOx as well as the NO2:NO ratio in the EmChem19 mechanism has been added to the revised manuscript.

4. What molecular weight is assigned to NMVOC, what is the species partition assumed in VOC? Or are emissions given in Tg(C) yr-1?

NMVOC have no particular molecular weight assigned to them. In reply to reviewer 1 as well as the above comment, additional information about the EmChem19 chemical mechanism has been added to the model description. The latter also contains information about the VOC emission splits in the EMEP model.

5. How are these emissions extrapolated into the future?

Forest fire emissions for each of the simulation years (2013 to 2017) are the same for all simulations, and its emissions are therefore assumed to remain effectively unchanged in the future. This is clarified in the revised manuscript text.

6. how consistent is this scenario in view of recent political changes (e.g., Brexit, war in Ukraine, COVID, etc.)? Please comment.

The baseline used in this work does not include any recent shock events (e.g., COVID-19). A statement on the scenarios not including recent shock events has been added to the revised manuscript test. What exactly the impact of these shock events is on the implementation of the already agreed upon EU27 legislative measures, is beyond our reach to quantify.

7. Turkey?

The spelling of Turkey as Türkiye follows the latest guidelines set by the United Nations.

8. It would be of great interest if the authors could comment on how these number for 2050 compares to present-day emissions from permafrost soils. What drives the changes between now and 2050? How much confidence do we have in those predictions?

As described in detail in the cited work of Scheider et al. (2012), permafrost in the MAGICC7 model is assumed, or tuned, to start thawing after 1 degree of global mean warming. As a result, modeled permafrost $CH_4$ emissions in our 2015 baseline scenario year are relatively small (600-ensemble mean of 0.73 Tg yr-1 and 5-95% range of 0.07-2.31 Tg yr-1).

While the permafrost module depends on freezing rates, amplification of permafrost area warming over global mean warming (polar amplification), turnover time of aerobic mineral soil fractions, and other factors, ultimately its parameters are driven by global mean temperature change (Schneider et al., 2012, Table 1). However, as noted by Schneider et al. (2012), at this time, process-based models constrained by observations are needed to better quantify permafrost-carbon and other permafrost feedbacks. In our view, this makes the results of the permafrost module in MAGICC7 a current best-guess indication of future permafrost $CH_4$ emissions.

The text in Section 3 has been updated to reflect some of the discussion points brought up in response to this comment. We have also included the 600-ensemble 5-95% range of the permafrost emissions by 2050 (0.54 – 11.22 Tg yr-1 for the CLE scenario), rather than referring only to its ensemble mean.

9. please explain the acronym

We have added Meteorological Synthesising Centre – West (MSC-W).

10. Is this enough given that the whole atmosphere methane lifetime is about 9.5 years and the perturbation lifetime is around 12 years? How far from equilibrium is methane in those simulations? Is the short methane lifetime compensated by setting methane concentrations individually to their intended concentration levels for each of the simulations prior to spin-up?

As described also in more detail in response to RC1 and comment 4 of RC2, it is clarified in the text that the EMEP model is run in a transient mode, though with methane concentrations prescribed (and fixed throughout the simulation period) at the start of each simulation. While the methane concentrations are therefore not transient (in contrast to the other chemistry; also that which depends on CH4), the prescribed CH4 concentrations are calculated using the transient MAGICC7 model. In effect carrying over the transient results of the MAGICC7 to the non-transient CH4 concentrations in the EMEP model.

11. It is not clear to me if the model is run in time-slice mode or transient mode. The scenarios would indicate transient, the constant methane concentration however suggests time-slice mode.

We hope to have addressed this comment in our reply to the above comment.

12. I am not sure I fully understand the concept of cascade plots. In the bars past 2050 CLE it seems that the bars between 2050 CLE and 2050 MFR (for instance) correspond to the reduction in MDA8 due to each contributing factor.

But that is not the case between 2015 and 2050CLE. I presume that NOx/VOC/NMVOC lead to a reduction while CH4 leads to an increase. So the direction is different. Perhaps one could indicate the direction of the impact with a small arrow to make the plot easier to read?

Indeed the direction is different for CH4 going from 2015 to 2050 CLE, as background CH4 concentrations increase in the CLE scenario.

Adding arrows for each of the NOx/VOC/NMVOC and CH4 steps was found to introduce considerable clutter. However, adding downward and upward arrows only from the baseline 2015 to 2050 CLE largely avoids this, while still serving as an illustration of the concept behind the cascade plot. By adding these we hope that the concept behind the figures is more easily understood, with the arrows also being referred to in the text. To make the arrows more visible (in terms of their size), we have also slightly adjusted the y-axis range in Fig. 3.

13. This implies strict or near linearity in the impact of ozone precursors on MDA8, correct?

It implies linearity (or near-linearity) in the impact on ozone precursors from 1) emission reductions in the EMEP region, 2) emission reductions in the ROW region, 3) methane changes.

For example, the '2015 Base to 2050 MFR' scenario is constructed using the '2015 Base to 2050 CLE' and the '2050 CLE to 2050 MFR' scenarios from Table 2. This then assumes that the impact of EMEP region emissions going from 2015 baseline to 2050 CLE and from 2050 CLE to 2050 MFR are additive, such that a net impact of EMEP region emission changes from 2015 baseline to 2050 MFR can be calculated from the sum of their impacts. However, in the 2015 baseline to 2050 CLE emission change, ROW emissions were at 2015 baseline levels, whereas in the 2050 CLE to 2050 MFR step, ROW emissions were at 2050 CLE levels. If the ROW emissions would affect the impact of EMEP region emission changes on surface ozone, the assumed (near-)linearity would not work, and we would not be able to construct the scenarios based on different sets of simulations. However, our modeling work (not explicitly shown) suggests that the impact of ROW emissions on the ozone reductions achieved by EMEP region emission reductions is limited, at least in the chemical regimes relevant to the current work. For example, diagnostic simulations found practically identical results between reducing ROW first and then EMEP emissions, versus reducing EMEP emissions first and then ROW.

Similarly, Section 6.1 argues that the impact of CH4 changes are also not particularly dependent on the choice of background emissions (at least not to the extent that it would affect the results of Table 3), such that the method of constructing composite scenarios based on different scenario simulations works. We note that the local ozone response to emission reductions (e.g., for NOx) in the EMEP region can still be highly non-linear for the above to hold. (with ozone produced in the ROW region contributing predominantly to background ozone)

14. I would not call a 10% increase in the total "small".

This was indeed poorly worded, and was meant to more strongly reflect that the change is expected to be smaller by 2050 than by 2100, and therefore comparatively small relative to the baseline natural emissions estimated for 2015. However, part of the increase calculated by Guo et al. (2023) is attributed to permafrost thawing, which is included as a factor in the MAGICC7 calculations. The revised manuscript reference to the latter as well as a slight rewording.

15. Perhaps I am missing the point here, but I am not quite sure how this comparison works. It seems to me that the two setups compared are considerably different and the comparable results are obtained more by confidence. How similar or different is the setup used by Belis and Van Dingenen to the one used in this study?

The point of this comparison to others is to provide context to our results by comparing them to what has been found before. The cited work of Belis and Van Dingenen (2023) use the TM5-FASt

model, based on pre-calculated transfer coefficients, or linear source-receptor coefficients (X amount of pollutant change in response to Y amount of emission change from country Z), to estimate changes in peak season MDA8 across the entire UNECE region in response to NOx/CO/NMVOC and CH4 changes, using similar CLE and MFR scenarios (but from the ECLIPSE v6b dataset).

In the revised manuscript, we added additional information regarding the difference in modeling setup compared to Belis and Van Dingenen (2023). We also extended the first sentence to provide motivation for the comparison/discussion of results.

**References**

Schneider von Deimling, T., Meinshausen, M., Levermann, A., Huber, V., Frieler, K., Lawrence, D. M., and Brovkin, V.: Estimating the near-surface permafrost-carbon feedback on global warming, Biogeosciences, 9, 649–665, https://doi.org/10.5194/bg-9-649-2012, 2012.

Belis, C. A. and Van Dingenen, R.: Air quality and related health impact in the UNECE region: source attribution and scenario analysis, Atmos. Chem. Phys., 23, 8225–8240, https://doi.org/10.5194/acp-23-8225-2023, 2023.

Guo, J., Feng, H., Peng, C., Chen, H., Xu, X., Ma, X., et al. (2023). Global climate change increases terrestrial soil CH$_4$ emissions. *Global Biogeochemical Cycles*, 37, e2021GB007255. https://doi.org/10.1029/2021GB007255